# FlexDrive: Toward Trajectory Flexibility in Driving Scene Reconstruction and Rendering

## Abstract

Driving scene reconstruction and rendering have advanced significantly using the 3D Gaussian Splatting. However, most prior research has focused on the rendering quality along a pre-recorded vehicle path and struggles to generalize to out-of-path viewpoints, which is caused by the lack of high-quality supervision in those out-of-path views. To address this issue, we introduce an Inverse View Warping technique to create compact and high-quality images as supervision for the reconstruction of the out-of-path views, enabling high-quality rendering results for those views. For accurate and robust inverse view warping, a depth bootstrap strategy is proposed to obtain on-the-fly dense depth maps during the optimization process, overcoming the sparsity and incompleteness of LiDAR depth data. Our method achieves superior in-path and out-of-path reconstruction and rendering performance on the widely adopted Waymo Open dataset. In addition, a simulator-based benchmark is proposed to obtain the out-of-path ground truth and quantitatively evaluate the performance of out-of-path rendering, where our method outperforms previous methods by a significant margin.

## 1 Introduction

3D reconstruction in driving scenes is a cornerstone of a high-quality driving visual simulator. Leveraging NeRF (Mildenhall et al., 2021) and the emerging 3D Gaussian Splatting (Kerbl et al., 2023b), the community has made significant progresses (Zhou et al., 2024; Yang et al., 2023a; Guo et al., 2023; Yan et al., 2024; Chen et al., 2023; Liu et al., 2023; Lu et al., 2023) in this area, possessing impressive rendering quality in the pre-recorded driving trajectories.

However, a significant issue hinders the current methods from being used in a practical simulator: the rendering quality declines significantly when the viewpoint deviates from the vehicle's trajectories for data collection. Fig. 1 demonstrates this issue. The essential cause for this issue is the unavailability of ground-truth visual observations from out-of-path viewpoints in driving scenes (Sun et al., 2020; Caesar et al., 2019; Geiger et al., 2013), where only pre-recorded images along a single-pass driving trajectory are available.

To address this issue, UniSim (Yang et al., 2023b) first introduces the concept of "lane shift" in their driving simulator, where they leverage GAN-generated supervision to refine the rendering quality of out-of-trajectory viewpoints. LidaRF (Sun et al., 2024) proposed to warp colors from in-path views to the target out-of-path views through the sparse LiDAR points, creating pseudo ground truth of the out-of-path views. However, due to the sparsity of LiDAR points and occlusion in the target view, the pseudo ground truth is usually broken and irregular, having quite different appearances from the real images captured by cameras. These limitations raise a natural question: *can we create a regular and complete pseudo ground truth for reconstructing the out-of-path views?*

We propose *Inverse View Warping* (IVW) to solve this challenge. Intuitively, IVW is the inverse process of the aforementioned color warping method. Considering an in-path view $A$ and an adjacent out-of-path view $B$, all content of $A$ can be captured at viewpoint $B$ if we omit the slight color change and occlusions caused by their different view direction. *Inverse View Warping* (IVW) tries to render the complete content of $A$ at the viewpoint $B$. In this circumstance, we can directly use $A$ as the ground truth to reconstruct the out-of-path view B. Specifically, to achieve this warping, we first unproject the pixels of view $A$ into 3D points and project those points into view $B$, forming a *warped ray map*. We then perform *occlusion-aware rasterization* according to the warped ray

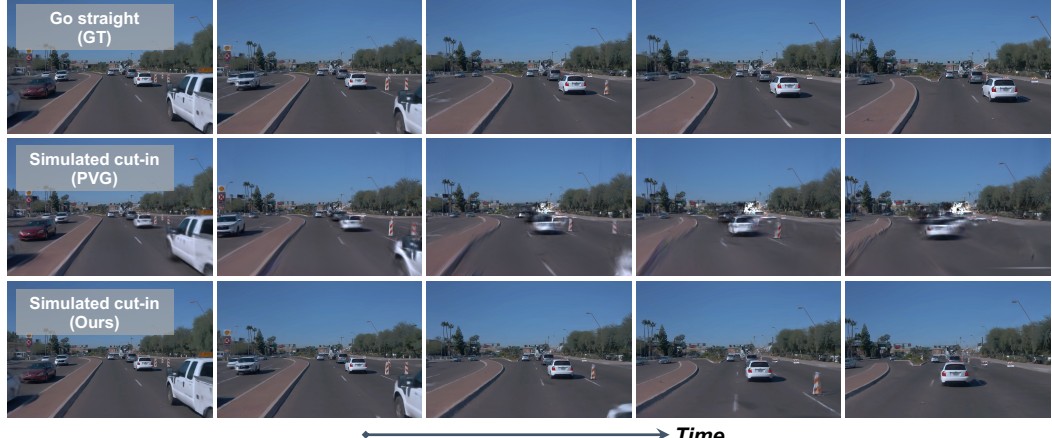

Figure 1: We simulate a cut-in case in a high-speed scenario, which is a typical functionality in driving simulators. The representative method PVG (Chen et al., 2023) fails after the lane change. We provide more video demonstrations in the attached supplementary materials.

map to obtain a warped rendering output. Finally, we *rearrange* the warped rendering results into a regular image, which should have the same appearance as the in-path counterpart. Thus we can supervise the rearranged rendering results using the in-path counterpart. Furthermore, this IVW technique necessitates accurate depth for point unprojection. To this end, we propose a novel Depth Bootstrap (DB) strategy to periodically refine the depth of the Gaussian field, leading to dense and accurate depth maps to support the IVW technique. The combination of IVW and DB leads to our overall framework FlexDrive.

Another hindrance to our goal is the lack of out-of-path ground truth for reliable evaluation. To address this, we turn to driving simulators where free-viewpoint ground truth images can be easily obtained. We build a benchmark based on the popular open-sourced CARLA simulator.

In summary, our contribution comes in four folds:

1. We propose Inverse View Warping, which creates high-quality supervision for out-of-path viewpoints in street scenes, significantly improving reconstruction quality from these novel viewpoints.

2. We propose a novel depth bootstrapping strategy to obtain a dense and accurate depth map, enabling more robust Inverse View Warping.

3. We build a new novel view synthesis benchmark upon the CARLA simulator to evaluate the out-of-path views.

4. In addition to competitive rendering quality in traditional in-path views, our method achieves superior performance in the out-of-path views, validated by quantitative and qualitative results in Waymo dataset and our proposed benchmark.

## 2 RELATED WORK

**3D Gaussian Splatting** 3D Gaussian Splatting (3DGS) (Kerbl et al., 2023a) have gained significant progress in scene modeling and rendering. While the original 3DGS model focuses on representing static scenes, several researchers have adapted it for dynamic objects and environments. (Yang et al., 2024; Wu et al., 2024; Huang et al., 2024) establishes dynamic Gaussian fields by introducing additional neural networks into the point clouds based on 3D Gaussian fields. Another group of researchers (Zhou et al., 2024; Yan et al., 2024) approaches this problem by developing 3D Gaussian fields which are naturally dynamic. However, the existing approaches are constrained as they can model only the in-path views scenes. Our work extends the reconstruction from in-path views to more flexible rendering locations which truly enables the simulation of autonomous driving tasks.

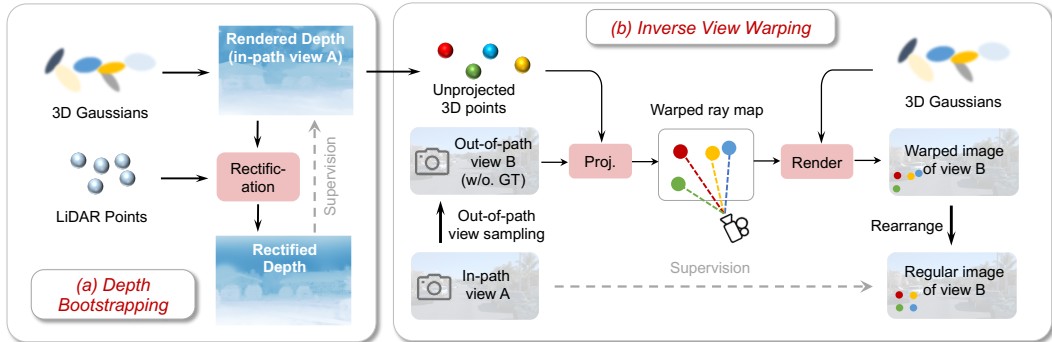

Figure 2: The main framework of FlexDrive, we provide high-quality supervision for out-of-path views through Inverse View Warping technique (IVW). To facilitate IVW, we propose Depth Bootstrapping (DB) to guarantee an accurate and dense depth map.

**3DGS in Autonomous Driving Simulation** Great efforts have been made to achieve higher reconstructing quality for autonomous driving scenes. Such reconstruction is essential for creating an autonomous driving environment. Although simulation environments such as CARLA (Dosovitskiy et al., 2017), and AirSim (Shah et al., 2018) exist, they require significant manual effort to create virtual environments and often lack realism in the generated data. A large number of studies have been devoted to this area (Cheng et al., 2022; Liu et al., 2023; Lu et al., 2023; Ost et al., 2021; Rematas et al., 2022; Tancik et al., 2022; Tonderski et al., 2024). These methods primarily concentrate on altering the autonomous driving scene along the data collection trajectory. For example, they can modify the lanes of neighboring cars or remove specific objects. However, simulating an autonomous driving scenario requires more than just these adjustments. The simulation environment must also accommodate maneuvers such as cut-ins, parallel parking, and turns. Achieving this necessitates flexible rendering capabilities, which have not been thoroughly explored in previous research.

## 3 METHOD

In this subsection, we first offer an overview of the proposed FlexDrive. Its overall architecture is demonstrated in Fig. 2, which has two major components including *Inverse View Warping* (IVW, Sec. 3.2) and *Depth Bootstrapping* (DB, Sec. 3.1). IVW creates high-quality visual supervision for training and improving the rendering at out-of-path virtual viewpoints. Since IVW relies on depth estimation, DB provides accurate and dense depth maps to enhance the IVW. Furthermore, we also improve dynamic object modeling (Sec. 3.3) to make the FlexDrive better support the reconstruction of dynamic scenes in out-of-path viewpoints. We summarize our optimization objectives in Sec. 3.4.

### 3.1 DEPTH BOOTSTRAPPING

Depth Bootstrapping leverages sparse LiDAR information to repeatedly rectify the dense depth map rendered from current reconstructed 3D Gaussians. We have two steps in Depth Bootstrapping: *Sparse Depth Initialization* and *Dense Depth Rectification*. The first step accumulates multi-frame LiDAR points and projects them into a training view to initialize the sparse depth map. The second step adopts an efficient linear optimization to minimize the gap between the rendered dense depth map and the sparse depth map.

**Sparse Depth Initialization** We first transform the 3D sparse LiDAR points into the 2D pixel plane of each in-path training view. For a view at time step $t$, we first accumulate the 3D LiDAR points in multiple frames (30 frames in our experiments) $[t, t + T]$ into frame $t$ with the provided LiDAR poses. For dynamic objects, we leverage their bounding boxes to move the in-box points to the corresponding positions in frame $t$. Although this transformation process is straightforward, there are two challenges: (1) Multiple points may be projected to the same image coordinates; (2) The points of occluded objects could penetrate the occluder due to the sparsity of LiDAR points and be mistakenly projected into the 2D pixel plane.

To address the two challenges, we propose several simple yet effective rules. Let $p_k$ denote the $k$-th LiDAR point, and its corresponding image coordinates are denoted as $i_k$. $\tau(k)$ and $d(k)$ stands for the timestamp and depth of the $k$-th point, respectively. We then use the following rules to select a subset of these points to build the sparse depth map.

1. If the depth of point $i_k$ deviates from the current rendered depth from 3D Gaussians[1] over a given threshold (e.g., $5\%$ current depth), the point is removed.

2. If point $i_{k_1}$ and point $i_{k_2}$ occupy the same pixel position with $\tau(k_1) < \tau(k_2)$, we keep $i_{k_1}$ and remove $i_{k_2}$.

3. If point $i_{k_1}$ and point $i_{k_2}$ occupy the same pixel position with depth $d(k_1) < d(k_2)$, we keep $i_{k_1}$ and remove $i_{k_2}$.

Intuitively, rule (1) indicates we only utilize the relatively accurate sparse depth and rule out the occluded points. Rule (2) and (3) indicate we prefer the "early appeared" points and closer points.

**Dense Depth Rectification**   Although the sparse depth map is relatively accurate, it only occupies a very small portion of the whole image plane, leading to several problems. (1) The supervision for Gaussian depth is sparse and makes it hard for the Gaussian field to render a smooth and continuous depth map. (2) Floaters, especially floaters in the regions that LiDAR cannot cover, cannot be effectively removed. (3) More importantly, with a sparse depth map, we can only build sparse visual supervision for the out-of-path viewpoint in the IVW (Sec. 3.2), making our method less effective.

To tackle these problems, we propose to densify the sparse depth map into a dense one. Our densification process is inspired by the observation that the rendered depth map is highly linear to the sparse depth map built from LiDAR, illustrated in Fig. 3. Thus, we rectify the rendered depth map into a more accurate one by solving a linear optimization problem. Since the rendered depth is naturally dense, in this sense, we "convert" the sparse depth map into a dense one.

Specifically, given the sparse depth $\mathcal{D}_s$ and the dense rendered depth $\mathcal{D}_r$, we find the best linear transform parameters that map $\mathcal{D}_r$ to $\mathcal{D}_s$. Then the rectified rendered depth map can be obtained as

$$\mathcal{D}'_r = a\mathcal{D}_r + b, \tag{1}$$

where mapping parameters $a, b$ minimize the optimization objective

$$L_{rect} = \sum_i \left\| \frac{a\mathcal{D}^i_s + b - \mathcal{D}^i_s}{\mathcal{D}^i_s} \right\|. \tag{2}$$

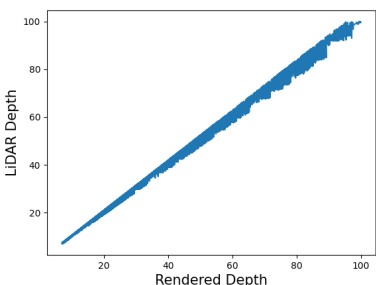

Here $i$ indexes the pixel location where sparse LiDAR depth is available. The parameters $a$ and $b$ can be efficiently solved with least squares method. Using the rectified depth $\mathcal{D}'_r$ as supervision, we then optimize the Gaussian field to make its depth more accurate. In this way, we obtain increasingly accurate dense rendered depth maps during the training process.

Figure 3: The strong linear prior between LiDAR depth and GS-rendered depth.

**Discussion**   Compared with direct LiDAR depth supervision as in (Chen et al., 2023; Sun et al., 2024), the superiority of our method stems from two aspects: (1) Only high-confident and reliable sparse depth is leveraged in the sparse map initialization step; (2) We utilize the strong linear prior, demonstrated by Fig. 3, to conduct depth bootstrapping and obtain dense and accurate depth maps.

### 3.2   INVERSE VIEW WARPING

With the accurate dense depth, we then conduct the proposed Inverse View Warping (IVW). The IVW procedure can be decomposed into three steps as shown by Fig. 4. We use $\mathcal{V}_{in}$ to denote the in-path view and $\mathcal{V}_{out}$ to denote a corresponding virtual out-of-path view, which is randomly sampled nearby the $\mathcal{V}_{in}$.

---

[1]The Gaussian field is warmed up for 5k iterations and has a relatively good initial depth.

**Warped Ray Map Generation** In this step, we first unproject the 2D pixel positions from the in-path view $\mathcal{V}_{in}$ back into the 3D space, using the rendered depth map. The obtained 3D points are then projected to the out-of-path view $\mathcal{V}_{out}$, resulting in a warped ray map as shown by Fig. 4. Since each ray corresponds to a pixel in $\mathcal{V}_{in}$, a warped image can also be obtained. Afterward, using the warped image to supervise the out-of-path view $\mathcal{V}_{out}$ seems a straightforward solution. However, such a solution is suboptimal because it may contain wrong colors due to the neglect of potential occlusion in $\mathcal{V}_{out}$. To address this challenge, we propose the following *Occlusion-aware Rasterization* technique.

**Occlusion-aware Rasterization** For each ray, we first sort the 3D Gaussian primitives along the ray according to their depth. We then adopt an alpha-blending process within a limited depth range, where the original alpha-blending process is modified to

$$C = \sum_{i=1}^{N} \mathbb{I}(d_i > \beta d_0)\alpha_i \prod_{j=1}^{i-1}(1-\alpha_j)c_i, \tag{3}$$

where $d_i$ is the depth of the $i$-th Gaussian primitive and $d_0$ is the depth of the unprojected point. Eq. (3) indicates that only primitives with a depth

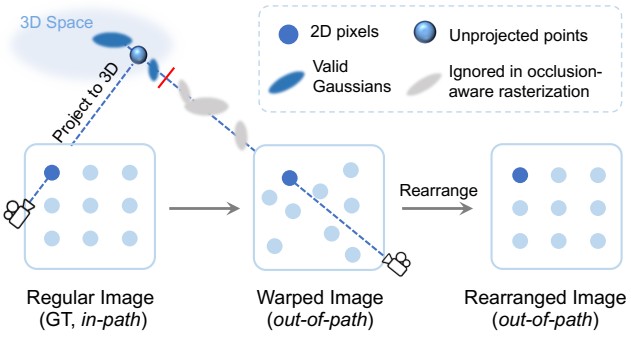

Figure 4: A demonstration for the Inverse Views Warping.

larger than $\beta d_0$ are involved in the alpha-blending process, illustrated by Fig. 4. Here we introduce $\beta$, a coefficient slightly smaller than 1, to take the thickness of Gaussian primitives into account, which avoids mistakenly neglecting the Gaussian primitives near the unprojected 3D points. In this way, even if some regions in $\mathcal{V}_{in}$ are occluded from $\mathcal{V}_{out}$, we can still provide accurate visual supervision for $\mathcal{V}_{out}$.

**Pixel Rearrangement** The output of occlusion-aware rasterization is a warped image, which has a quite different appearance from the regular image at the in-path view, demonstrated by Fig. 5 (b-c). Thus, we cannot employ the region-level perceptual loss such as SSIM and LPIPS with the warped images. To address this issue, we rearrange the rendered pixels in $\mathcal{V}_{out}$ and recover their relative spatial orders in $\mathcal{V}_{in}$. In this way, the rendering result in $\mathcal{V}_{out}$ is expected to be the same as the ground truth image in $\mathcal{V}_{in}$ if the Gaussian field is sufficiently optimized, demonstrated by Fig. 5 (d). We then can use the ground truth image in $\mathcal{V}_{in}$ as the supervision for $\mathcal{V}_{out}$.

### 3.3 CONSTRAINED DYNAMIC OBJECT MODELING

Dynamic objects are important components in driving scenarios. Although previous research (Huang et al., 2024; Yan et al., 2024) has made notable success in modeling them at in-path vies, we notice that these methods usually result in tailed floaters around the dynamic objects. The

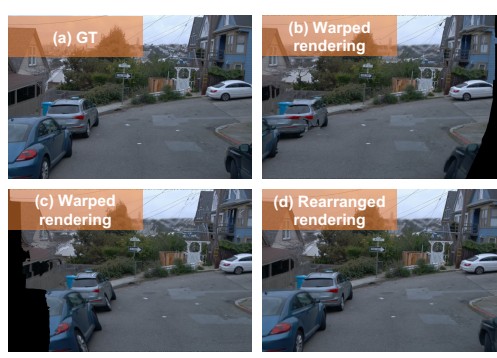

Figure 5: Example of rearrangement. (a) is the ground truth in-path view. (b) and (c) are the warped rendering results in out-of-path views (right and left shifted, respectively). Note we ignore those rendered pixels out of image boundaries in this illustration. However, in practice, we still keep the out-of-boundary pixels and rearrange them. (d) is the rearranged rendering results in out-of-path views, which is almost the same as the GT image (a).

tailed floaters severely lower the rendering quality of out-of-path viewpoints. To alleviate this issue, we propose to use a constrained modeling strategy for dynamic objects.

Following (Zhou et al., 2024; Yan et al., 2024), we represent each dynamic object with a separate 3D Gaussian field. However, different from previous strategies, each dynamic object is constrained in a bounding box in our framework. Let $(x_o, y_o, z_o)$ be the logistic coordinates of a Gaussian primitive, we convert them into Euclidean coordinates following

$$\begin{bmatrix} x_t \\ y_t \\ z_t \end{bmatrix} = \begin{bmatrix} l(\sigma(x_o) - 0.5) \\ w(\sigma(y_o) - 0.5) \\ h(\sigma(z_o) - 0.5) \end{bmatrix}. \tag{4}$$

The converted Euclidean coordinates are further transformed into the world coordinates by a trainable bounding box pose (details can be found in the appendix).

### 3.4 Loss Functions

**RGB Loss**   We employ the original RGB loss setting for both in-path views and out-of-path views. They are both supervised by a mixture of $L_1$ loss and SSIM loss. The overall loss of RGB part can be formulated as

$$L_{RGB} = L_1^{in} + L_1^{out} + \alpha(L_{SSIM}^{in} + L_{SSIM}^{out}), \tag{5}$$

where the superscript $in$ and $out$ stands for in-path views and out-of-path views.

**Depth Loss**   We categorize depth supervision into the *near* and *far* regions according to the maximum LiDAR perception range. Let $d_i$ and $\hat{d}_i$ be the depth of $i$-th pixel in the rendered depth map and the rectified depth map (Eq.( 1)), respectively. Then the near-region depth supervision is defined as

$$L_{depth}^{near} = \frac{1}{N_{near}} \sum_{i=1}^{hw} \mathbb{I}(d_i < d_{max}) \| \frac{d_i - \hat{d}_i}{\hat{d}_i + \epsilon} \|_1, \tag{6}$$

where $d_{max}$ is the maximum LiDAR perception range and $N_{near}$ is the number of near-region pixels in the depth map. For the far-region depth loss $L_{depth}^{far}$, we directly adopt the ranking loss in (Wang et al., 2023).

The total loss function of FlexDrive can be formulated as

$$L = \lambda_1 L_{RGB} + \lambda_2 L_{depth}^{near} + \lambda_3 L_{depth}^{far}. \tag{7}$$

## 4 Experiments

In this section, we evaluate our method on the real-world Waymo dataset and the proposed CARLA-based dataset to accurately evaluate the performance of both in-path setting and out-of-path setting.

### 4.1 Experiment Setup

**Waymo-based in-path Benchmark**   We first follow the conventional practices (Chen et al., 2023; Yan et al., 2024) to evaluate the performance of the proposed method in the widely used Waymo Open Dataset (WOD). Similar to PVG (Chen et al., 2023), we conduct our experiments on both dynamic and static split of the Waymo dataset and use the three front cameras. Since there are no ground-truth images of out-of-path views, we mainly focus on the qualitative results for the out-of-path views in the Waymo Open dataset. Additionally, we report FID scores of the out-of-path views as an intuitive but potentially inaccurate quality indicator.

**CARLA-based out-of-path Benchmark**   Since the distribution-based FID is a rough metric for the rendering quality, we propose a new benchmark based on the CARLA simulator (Dosovitskiy et al., 2017), where the ground-truth images of out-of-path views can be easily obtained. The CARLA-based benchmark is set up with a sensor layout similar to the Waymo dataset. Specifically, we mount five cameras on the data-collection vehicle. Three of them are placed on the top of the vehicle to record the in-path training data. The other two cameras are shifted horizontally three meters away from the moving path to collect the out-of-path ground truth images for evaluation. A 150-meter range 128-channel LiDAR is mounted on top of the moving vehicle.

Table 1: The performance comparison on the Waymo static scenes. We report PSNR and SSIM for the in-path setting and FID for the out-of-path setting. The results are obtained with the default training iterations in the official code.

| Model Setting | PSNR ↑ | SSIM ↑ | FID@1 meters ↓ | FID@2 meters↓ |
|---|---|---|---|---|
| 3D GS (Kerbl et al., 2023a) | 29.40 | 0.892 | 85.22 | 120.34 |
| EmerNeRF (Yang et al., 2023a) | 30.15 | 0.828 | 65.05 | 82.42 |
| PVG (Chen et al., 2023) | 30.13 | 0.877 | 75.97 | 99.11 |
| LidaRF (Sun et al., 2024) | 29.72 | 0.889 | 69.28 | 95.46 |
| StreetGaussian (Yan et al., 2024) | 31.35 | 0.911 | 72.03 | 95.34 |
| FlexDrive (ours) | 30.00 | 0.878 | 62.03 | 86.05 |

**Training Scheme**  Our training process can be divided into three stages: (1) the warm-up stage, (2) the bootstrapping stage, and (3) the out-of-path training stage. During the warm-up stage, we initialize the 3D Gaussian primitives using multi-frame LiDAR points and conduct training in the in-path views without Gaussian densification. Single-frame sparse LiDAR supervision is directly employed in the in-path views, following (Chen et al., 2023; Sun et al., 2024). In the second stage, we enable the proposed depth bootstrapping and the densification strategy in the original 3DGS. The parameters in the linear transformation (Eq. (1)) are solved by the least squares method. Finally, we begin the out-of-path training stage. In this stage, we sample an in-path view and randomly generate a nearby out-of-path view for each iteration, as Fig. 2 (b) shows. The rendering results in the in-path view and out-of-path view are supervised by the in-path ground truth images and virtual ground truth image created in Sec. 3.2. The three stages take 5k, 15k, and 10k, respectively. More detailed hyperparameters can be found in the appendix.

**Compared Methods**  In our experiments, we compare our method with both NeRF-based and GS-based baselines. Specifically, we adopt five typical methods for comparison, including EmerN-eRF (Yang et al., 2023a), LidaRF (Sun et al., 2024), 3DGS (Kerbl et al., 2023a), StreetGaussian (Yan et al., 2024), and PVG (Chen et al., 2023). The NeRF-based LidaRF is the most recent state-of-the-art method for out-of-path rendering. However, this method has not been open-sourced, thus we re-implement LidaRF by ourselves. We further transfer the techniques in LidaRF to 3D Gaussian Splatting, resulting in a LidaRF-GS.

## 4.2 RESULTS ON WAYMO DATASET

**Quantitative Results**  We first report the quantitative results in both in-path and out-of-path settings on the Waymo Open dataset. For in-path rendering, conventional metrics PSRN and SSIM are reported. For the out-of-path rendering, we report FID scores as a rough quality indicator since the ground truth images of out-of-path viewpoints are not available. The source distribution used in FID is the in-path ground truth images, and the target distributions are sampled at poses laterally shifted 1 meters and 2 meters away from the vehicle path, respectively. As shown in Table 1 and Table 2, the proposed FlexDrive achieves comparable performance with the compared methods on the in-path rendering task. When the viewpoints shift away from the in-vehicle path, FlexDrive also achieves relatively good FID scores. However, we emphasize that the distribution-based FID score is not a reliable criterion for rendering quality evaluation because it only indicates the overall distribution similarity instead of the detailed rendering quality.

**Qualitative Results**  We further provide the qualitative results of out-of-path rendering as Fig. 6 shows, where FlexDrive demonstrates significant rendering quality in the out-of-path setting.

## 4.3 RESULTS ON CARLA-BASED DATASET

To accurately evaluate the performance on the out-of-path viewpoints, we further build a new benchmark upon the CARLA simulator where the ground truth images of out-of-path viewpoints are available. The detailed setting of this benchmark is presented in Sec. 4.1. In Table 3, our method largely outperforms the previous street scene reconstruction method in the out-of-path rendering. Notably,

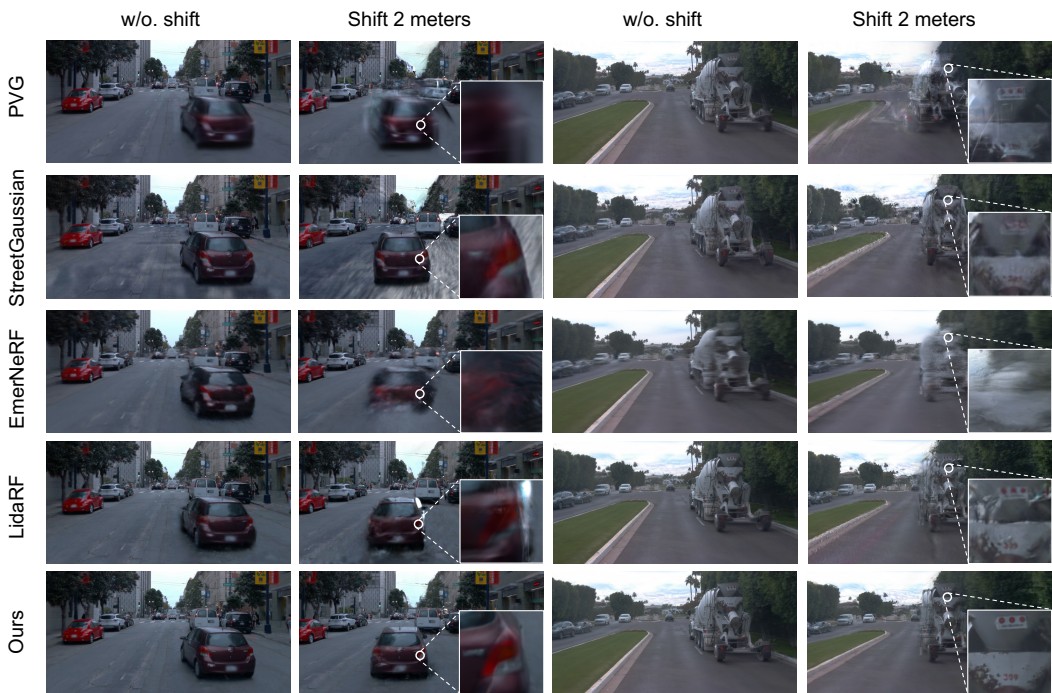

Figure 6: Qualitative comparison. We provide more video demonstrations in the attached supplementary materials.

Table 2: Performance comparison on the Waymo dynamic scenes. We report PSNR and SSIM for the in-path setting and FID for the out-of-path setting. The results are obtained with the default training iterations in the official code.

| Model Setting | PSNR ↑ | SSIM ↑ | FID@1 meters ↓ | FID@2 meters↓ |
|---|---|---|---|---|
| 3D GS (Kerbl et al., 2023a) | 28.40 | 0.869 | 100.01 | 126.77 |
| EmerNeRF (Yang et al., 2023a) | 28.21 | 0.800 | 83.53 | 106.6 |
| PVG (Chen et al., 2023) | 29.77 | 0.872 | 52.54 | 81.76 |
| LidaRF (Sun et al., 2024) | 30.21 | 0.878 | 59.26 | 83.41 |
| StreetGaussian (Yan et al., 2024) | 30.73 | 0.883 | 78.23 | 110.6 |
| FlexDrive (Ours) | 29.92 | 0.886 | 58.12 | 85.06 |

Table 3: CARLA-based out-of-path evaluation. We report the results with the default training iterations in their official code.

| Model Setting | PSNR ↑ | SSIM ↑ | LPIPS ↓ |
|---|---|---|---|
| 3D GS (Kerbl et al., 2023a) | 18.90 | 0.701 | 0.565 |
| EmerNeRF (Yang et al., 2023a) | 21.18 | 0.788 | 0.463 |
| PVG (Chen et al., 2023) | 21.65 | 0.753 | 0.444 |
| LidaRF (Sun et al., 2024) | 24.84 | 0.852 | 0.402 |
| StreetGaussian (Yan et al., 2024) | 24.68 | 0.876 | 0.411 |
| LidaRF-GS | 24.79 | 0.842 | 0.410 |
| FlexDrive (ours) | 26.23 | 0.877 | 0.372 |

FlexDrive achieves a significant performance gain in terms of PSNR compared with the LidaRF, which also focuses on the out-of-path setting.

Table 4: The overall ablation of our proposed techniques. All models are evaluated in the CARLA-based out-of-path setting. The LidaRF-GS is an adaption of LidaRF (Sun et al., 2024) to 3DGS, serving as our baseline.

| Model Setting | PSNR ↑ | SSIM ↑ | LPIPS ↓ |
|---|---|---|---|
| LidaRF-GS | 24.79 | 0.842 | 0.410 |
| LidaRF-GS + DB | 25.57 | 0.866 | 0.381 |
| LidaRF-GS + IVW | 25.71 | 0.876 | 0.380 |
| LidaRF-GS + DB + IVW (full model) | 26.23 | 0.877 | 0.372 |

Table 5: Effectiveness of the occlusion-aware rasterization. $\beta = 0$ means that we do not handle the occlusion problem.

| $\beta$ | PSNR (single scene) |
|---|---|
| 0.95 | 32.23 |
| 0.8 | 30.90 |
| 0.5 | 29.65 |
| 0 | 20.12 |

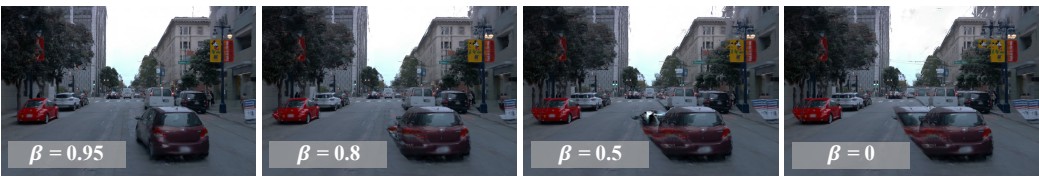

Figure 8: The rearranged rendering results in out-of-path views with different $\beta$ (Eq. (3)). Without the occlusion mechanism ($\beta = 0$), we have incorrect supervision for the out-of-path views.

## 4.4 ABLATION STUDY

In this subsection, we study the impact of each of our proposed modules. We first conduct an overall ablation for all the proposed modules and then delve into their detailed designs.

**Overall Ablation**  We first adopt NeRF-based LidaRF (Sun et al., 2024) to 3D Gaussian Splatting as our baseline named LidaRF-GS for a step-by-step ablation. We then add the proposed depth bootstrapping and inverse view warping step-by-step to the LidaRF-GS baseline to reveal the performance roadmap. All models are trained with the three stages introduced in Sec. 4.1. We use 400k initial points for all ablation settings. The maximum iteration number is set to 35k. The results in Table 4 demonstrate that our proposed techniques are all effective and the depth bootstrapping technique indeed enhances the inverse view warping.

**Depth Bootstrapping**  Depth noise and missing are inevitable in real-world datasets and it may lead to significant errors in the Inverse View Warping module. Fortunately, in FlexDrive, the depth bootstrapping module largely alleviates this issue. Here we use a scene in the real-world Waymo dataset to reveal the efficacy of this module. Fig. 7 illustrates the rearranged rendering results (similar to Fig. 5 (d)) at out-of-path views with and without depth bootstrapping. As can be seen, DB could effectively enhance the rendering quality, especially for those far regions uncovered by LiDAR. This is because those far regions can also be rectified by Eq. (1).

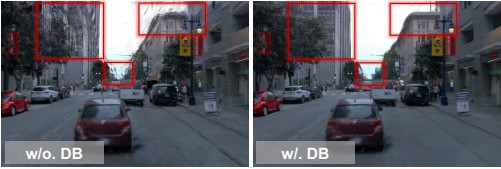

Figure 7: Effectiveness of Depth Bootstrapping.

**Ccclusion-aware Rasterization in IVW**  Occlusion is a key challenge in our inverse view warping strategy. We introduce $\beta$ to employ a depth range limitation in the alpha-blending process in Eq. (3). Here we study how this parameter impacts the inverse view warping strategy. As shown in Fig. 8, without handling the occlusion ($\beta = 0$), the rearranged results in the out-of-path view are not similar to the in-path ground truth, causing incorrect supervision signals. Such incorrect supervision signals not only reduce the out-of-path rendering quality but also affect the in-path rendering quality since the Gaussian primitives are shared. So we further provide the in-path quantitative results corresponding to Fig. 8, shown in Table 5. The in-path performance has a dramatic drop without depth bootstrapping.

## 5 CONCLUSIONS

To summarize, the proposed FlexDrive introduces the Inverse View Warping and Depth Bootstrap strategy for enhancing the reconstruction quality of street scenes, particularly from out-of-path viewpoints. Furthermore, the development of a new benchmark using the CARLA simulator allows for comprehensive evaluation of out-of-path views. The results demonstrate that not only does our method maintain competitive rendering quality in traditional in-path scenarios, but it also excels in out-of-path views. We provide both quantitative and qualitative analyses conducted on the Waymo dataset and our CARLA-based benchmark. This advancement opens new avenues for flexible rendering in reconstructed street scenes, in the future, we plan to combine our method with generative methods to allow completely free camera movement.

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

# A    APPENDIX

## A.1    FID SCORE IS NOT A RELIABLE CRITERIA FOR IMAGE SYNTHESIS

The Fréchet inception distance (FID) is a metric for measuring the distance between two distributions. With the normal distribution assumption, FID scores can be computed with the following formula:

$$FID = \|\mu_1 - \mu_2\| + \text{Tr}(\sigma_1 + \sigma_2 - 2\sqrt{\sigma_1 * \sigma_2}) \tag{8}$$

In our setting, we compute the FID scores between the in-vehicle path views $(\mu_1, \sigma_1)$ and the rendered shifted views $(\mu_2, \sigma_2)$. Because the shifted views are different from the in-vehicle path views by nature, thus $\mu_1 \neq \mu_2$ and $\sigma_1 \neq \sigma_2$. Now even if we simply shift the mean of $\mu_2$ to $\mu_1$, we can reduce the FID scores lower than the GT views. For this reason, we claim that FID score is not a good metric for the flexible rendering task.

## A.2    DETAIL OF CONSTRAINED DYNAMIC OBJECT MODELING

Let $(x_o, y_o, z_o)$ be the logistic coordinates of a Gaussian primitive, and $(l, w, h)$ be the length, width, and height of the bounding box containing this Gaussian primitive. We convert the logistic coordinates into Euclidean coordinates as following

$$\begin{bmatrix} x_t \\ y_t \\ z_t \end{bmatrix} = \begin{bmatrix} l(\sigma(x_o) - 0.5) \\ w(\sigma(y_o) - 0.5) \\ h(\sigma(z_o) - 0.5) \end{bmatrix}. \tag{9}$$

Next, we transform these local Euclidean coordinates into the world coordinates system. We introduce a sequence of trainable pose parameters $R_t$ and $T_t$. Here $R_t$ is a rotation matrix to transform the local system to the world system at time frame $t$, $T_t$ is the respective offset vector. Then, the world coordinates $(\bar{x}_t, \bar{y}_t, \bar{z}_t)$ can be computed as:

$$\begin{bmatrix} \bar{x}_t \\ \bar{y}_t \\ \bar{z}_t \end{bmatrix} = R_t \begin{bmatrix} x_t \\ y_t \\ z_t \end{bmatrix} + T_t. \tag{10}$$

## A.3    ALPHA BLENDED DEPTH

In our approach, depth prediction is crucial. To facilitate the original 3D Gaussian splatting field with our depth prediction, we render depth views in a normalized alpha-blending manner follows

$$D = \frac{1}{\sum_{i=1}^{N} T_i \alpha_i} \sum_{i=1}^{N} \alpha_i T_i d_i \text{ with } T_i = \prod_{j=1}^{i-1} (1 - \alpha_j), \tag{11}$$

Here $d_i$ is the depth of the 3D Gaussian center with respect to the current camera location. The blended depth can be directly viewed as a weighted average of Gaussian points' depth based on their importance.

## A.4    DETAILED TRAINING SETTINGS

In our experiments, we employed the Adam optimizer with a base learning rate of $2.5e - 3$. We focused on two primary hyperparameters relevant to our methods: the bootstrap update interval and the occlusion-aware Inverse View Warping. For the depth bootstrap strategy, we updated the accumulated sparse LiDAR map every 2 epochs, integrating 30 frames of LiDAR data into a single sparse depth map. In the Inverse View Warping module, we adopt $\beta = 0.95$.

