# OpenReview forum: "FlexDrive: Toward Trajectory Flexibility in Driving Scene Reconstruction and Rendering"
_ICLR.cc/2025/Conference — ICLR 2025 Conference Withdrawn Submission_

### Official Review · Reviewer_BX3v · 2024-10-28

**Soundness:** 3
**Presentation:** 3
**Contribution:** 2
**Rating:** 6
**Confidence:** 4

**Summary:**

The paper presents FlexDrive, a method that improves upon existing scene reconstruction techniques to allow for a more novel view synthesis (such as lane changes). The method first generates dense depth prediction for the input views by aligning the densely estimated depth with accumulated LiDAR points (whose rendering technique is also newly proposed to handle occlusions). To reach good generalization for novel views, the method uses the inverse view warping technique that warps the input views to the target views via the previously estimated depth. The wrapped images and depths are used to further supervise the reconstruction method to reach a decent novel view synthesis. The method is compared to many state-of-the-art approaches and demonstrates a better novel view synthesis results.

**Strengths:**

1. Compared to existing methods, the quality of the novel view synthesis looks good. This is well-demonstrated through the paper, as well as the accompanying video.

2. A new synthetic CARLA benchmark is provided, allowing for evaluating novel-view synthesis in a more principled way. This complements the real-world Waymo dataset for a better understanding of the model's superiority.

3. The method is simple and straightforward to implement, allowing for easy re-implementation using state-of-the-art reconstruction frameworks (both 3DGS and NeRFs).

**Weaknesses:**

1. The method seems to be utilizing the 3D Gaussian representation with dynamic rigid nodes. It would be nice to add detailed descriptions of such a representation (including the initialization and the density control strategies). How does the method handle non-rigid moving objects such as pedestrians or people in wheelchairs?

2. The inverse depth warping method does not seem to be completely principled to me, since the quality of the novel view synthesis is dependent on the newly-added views. It would be nice to show the performance of the method when the supervision view misaligns with the target views. Also for extreme novel view synthesis where the model might need to halluciate occluded contents and the corresponding view-dependent effects, how could the method handle such situations?

3. Some of the necessary visualizations or analyses are missing. For depth bootstrapping, the paper proposes a new method of rasterizing LiDAR point clouds onto the images. How does this compare to a trivial rasterization of points (e.g. with Pulsar [a])? The dense depth rectification stage recovers a scalar value for the entire image, and how does the rectified depth align with the LiDAR projection? It would be nice to show more visualizations of the effectiveness of the two methods proposed in Sec 3.1.

4. Reference to [b] is missing.

[a] Lassner, Christoph, and Michael Zollhofer. "Pulsar: Efficient sphere-based neural rendering." Proceedings of the IEEE/CVF Conference on Computer Vision and Pattern Recognition. 2021.

[b] Chen, Ziyu, et al. "OmniRe: Omni Urban Scene Reconstruction." arXiv preprint arXiv:2408.16760 (2024).

**Questions:**

I have no outstanding questions to ask at the moment.

---

### Official Review · Reviewer_wZtQ · 2024-10-31

**Soundness:** 1
**Presentation:** 1
**Contribution:** 2
**Rating:** 3
**Confidence:** 5

**Summary:**

This paper introduces a 3D Gaussian Splatting (3DGS) method aimed at enhancing novel view synthesis for autonomous driving scenarios. The approach seeks to improve 3DGS rendering quality for viewpoints significantly different from the training perspectives by incorporating LiDAR data. The central idea is to utilize LiDAR depth to warp training images into these new viewpoints, generating additional pseudo-training images and thereby enhancing the robustness and rendering quality of 3DGS. However, the paper appears hastily prepared, with core technical contributions not clearly articulated and comparison methods possibly biased. Furthermore, the proposed inverse warping strategy has been presented in several previous studies, including the ICLR 2023 paper "UC-NeRF: Neural Radiance Field for Under-Calibrated Multi-View Cameras in Autonomous Driving."

**Strengths:**

1. The paper addresses a crucial challenge in neural rendering for autonomous driving, where biased input training images often limit effective free-viewpoint rendering.

2. Experimental results demonstrate the effectiveness of the inverse warping approach.

**Weaknesses:**

Here is a refined version:

1. **Presentation Quality:** The paper’s presentation needs substantial improvement. While the length of under 10 pages is acceptable, the core technical details are insufficiently explained in the main content, making it challenging to grasp how the proposed method functions.

2. **Dense Depth Rectification:** The “dense depth rectification” operation lacks clarity. It appears to assume a linear relationship between rendered depth values and LiDAR depth, which is somewhat questionable. If 3DGS accurately captures scene geometry, the rendered depth should ideally match the LiDAR depth. However, 3DGS may not capture the scene accurately in all regions, leading to errors in the rendered depth map that cannot be corrected by a simple linear transformation. This raises concerns regarding the effectiveness of using a global linear transformation for depth rectification. The authors should clarify this process and provide experimental evidence, such as visualizations or evaluations of rendered depth maps with and without rectification, to substantiate their approach.

3. **Inverse Warping and Pixel Rearrangement:** The paper provides limited information on the pixel rearrangement operation within the inverse warping process, making it difficult to understand its function. A more detailed explanation of this component is needed.

4. **Comparisons with Prior Works:** The comparisons to prior works may not be entirely fair. The proposed method uses both RGB images and LiDAR data as input, whereas compared methods, such as 3DGS, typically use only RGB images. It is unclear whether the prior works included LiDAR data for their comparisons. The authors should clarify this to ensure a fair evaluation.

5. **StreetGaussian Comparison:** StreetGaussian, a closely related work that also utilizes LiDAR data, appears to achieve better results than the proposed method. However, visual results of StreetGaussian are not included in the supplementary videos, which raises concerns about whether StreetGaussian may offer superior visual quality.

6. **Novelty and Prior Work:** The concept of inverse warping has been applied in previous research, including the ICLR paper *"UC-NeRF: Neural Radiance Field for Under-Calibrated Multi-View Cameras in Autonomous Driving."* The authors should discuss how their approach differs from existing work and highlight its novel aspects.

**Questions:**

1. How the pixel rearrangements works?
2. Is Eq.2 correct? Since there only D_s^i is involved?

---

### Official Review · Reviewer_TjVx · 2024-11-09

**Soundness:** 3
**Presentation:** 2
**Contribution:** 3
**Rating:** 5
**Confidence:** 4

**Summary:**

This paper introduces an Inverse View Warping method to address camera simulation challenges for out-of-path views. Unlike most existing methods focused on rendering along a pre-recorded vehicle path, this approach achieves high-quality renderings even when vehicle paths deviate significantly. The proposed Inverse View Warping method generates compact, high-quality images to supervise the reconstruction of out-of-path views, leveraging LiDAR depth from alternative viewpoints and employing a depth bootstrap strategy to refine depth rendering in these views. The proposed method achieves superior reconstruction and rendering performance for both in-path and out-of-path views.

**Strengths:**

* This paper addresses a key challenge in autonomous driving: simulating camera images along novel trajectories that deviate from a pre-recorded path, which enables safer evaluation and development of autonomous systems.
* The proposed Inverse View Warping method generates high-quality renderings to supervise the reconstruction of out-of-path views.
* This paper presents a straightforward and effective approach for obtaining sparse depth maps by aggregating LiDAR data, handling occlusions, and using this data to rectify the rendered dense depth maps.
* Additionally, the paper highlights limitations of the distribution-based FID metric for out-of-path evaluation, noting that it compares in-path ground truth images with images from novel trajectories, which may lead to inaccuracies. To address this, the authors introduce a new benchmark using the CARLA simulator for evaluating out-of-path views.

**Weaknesses:**

* In Occlusion-aware Rasterization, why are the Gaussians in front of the warped depth from other frames removed? Projected 3D points from other frames could be occluded in the target frame.
* I’m unclear on the pixel rearrangement process after occlusion-aware rasterization. Could the authors elaborate on this step?
* For dynamic objects, the authors use bounding boxes to transform in-box points to the target frames. What might occur if bounding box labels are inaccurate or if shape distortion arises from rolling shutter LiDAR? Could the proposed framework potentially learn label refinement or model shape distortion?
* The paper claims to achieve out-of-path camera simulation, yet it only evaluates lane shifts of 1 and 2 meters. I would like to see results in a more challenging setting involving completely new trajectories.
* Additionally, the authors might consider evaluating actor rotation or shifts. The CARLA-based benchmark proposed here could be very useful for assessing performance in these challenging settings.
* If possible, I’d like to see a comparison with UniSim or NeuRAD, as these methods incorporate designs to handle extrapolation.

**Questions:**

This paper addresses a significant problem in autonomous driving. I have outlined several questions for the authors above, and I am open to adjusting my rating based on their responses in the rebuttal.

---

### Note · Authors · 2024-11-14

I have read and agree with the venue's withdrawal policy on behalf of myself and my co-authors.